# Application of Two-Photon-Absorption Pulsed Laser for Single-Event-Effects Sensitivity Mapping Technology

**DOI:** 10.3390/ma12203411

**Published:** 2019-10-18

**Authors:** Cheng Gu, Rui Chen, George Belev, Shuting Shi, Haonan Tian, Issam Nofal, Li Chen

**Affiliations:** 1Electrical Computer Engineering, University of Saskatchewan, Engineering Building 57 Campus Dr, Saskatoon, SK S7N5A9, Canada; gucheng.alex@usask.ca (C.G.); sst1124@163.com (S.S.); hat541@mail.usask.ca (H.T.); lic900@mail.usask.ca (L.C.); 2Saskatchewan Structural Science Centre, University of Saskatchewan, Thorvaldson Building Office 192, 110 Science Place, Saskatoon, SK S7N5E2, Canada; gsb808@mail.usask.ca; 3iROC Technologies, 38000 Grenoble, France; issam.nofal@iroctech.com

**Keywords:** pulsed laser, single event effect, SRAM, two photon absorption, sensitive mapping

## Abstract

Single-event effects (SEEs) in integrated circuits and devices can be studied by utilizing ultra-fast pulsed laser system through Two Photon Absorption process. This paper presents technical ways to characterize key factors for laser based SEEs mapping testing system: output power from laser source, spot size focused by objective lens, opening window of Pockels cell, and calibration of injected laser energy. The laser based SEEs mapping testing system can work in a stable and controllable status by applying these methods. Furthermore, a sensitivity map of a Static Random Access Memory (SRAM) cell with a 65 nm technique node was created through the established laser system. The sensitivity map of the SRAM cell was compared to a map generated by a commercial simulation tool (TFIT), and the two matched well. In addition, experiments in this paper also provided energy distribution profile along Z axis that is the direction of the pulsed laser injection and threshold energy for different SRAM structures.

## 1. Introduction

Single Event Effects (SEEs) are electrical disturbance in Integrated Circuits (ICs) or analogue circuits when they are hit by ionizing particles through or near sensitive nodes (a small area inside IC that is sensitive to external electronic stimulation) [1,2]. When the ionizing particle passes through the semiconductor materials in the IC, a number of Electron-Hole pairs (EHPs) will be generated, which is the reason for the circuits disturbance.

A traditional testing method for SEEs is Heavy Ion (HI) or protons and neutrons testing, which uses an accelerator to provide particles with certain energy and ICs devices would be exposed to the ion beams for simulating the SEEs phenomenon in natural space [3,4]. Although HI testing is a direct way to study SEEs, it doesn’t provide enough temporal and spatial information to understand mechanisms responsible for SEEs [5].

Pulsed laser systems have been introduced and widely used for simulating Single Event Effects procedure by generating EHPs through photon-particle interaction [4,5,6]. For SRAM and Flip-Flop cells, laser systems can be used to find the sensitive area [7,8,9,10]. Due to the ability of changing focus length, a 3-D SEEs profile can be also achieved by laser systems [11,12]. Meanwhile, for the abilities of fast scanning speed and repeatability, the laser system can be further used to find sensitive area in the whole area of IC circuit or analogue circuit [13,14]. As a complementary SEEs testing tool for particle accelerators, the injected laser energy could be correlated with the Linear Energy Transfer (LET) value in particle accelerator testing, which helps us to build a solid relationship between these two testing methods [15,16].

From basic physical theory, if an EHP needs to be generated in semiconductors, an electron located in Valence band of a semiconductor should receive enough energy that shall be no less than the bandgap ($E_{g}$) of the material, so as to be excited up to the Conduct band. Based on Planck’s Equation (E=hν) and light speed Equation (c=λν), if we let the energy of a single photon be the bandgap of silicon (1.12 eV), we can calculate a threshold wavelength as 1108 nm. When the wavelength of injected laser light is shorter than this threshold value, one single photon can generate an EHP. This kind of absorption process is called Single-Photon Absorption (SPA); whereas, if the wavelength of injected laser is longer than this threshold value, an EHP can not be generated by a single injected photon. However, if we can specifically restrict the energy of one single photon of injected laser to be less than $E_{g}$, but greater than $E_{g}/2$, in the domain of non-linear optics, two photons can be absorbed simultaneously and generating an EHP in the material. We call this process as Two-Photon Absorption (TPA). The occurrence of TPA process is highly depended on the input laser irradiance based on non-linear optical theory, and can only occur in the focused zone [17,18,19].

The following equation is mostly used to describe the EHPs generation in a semiconductor material [20].
(1)$$\frac{dN(r,z)}{dt}=\frac{\alpha I(r,z)}{h\nu }+\frac{\beta_{2}I^{2}\left(r,z\right)}{2h\nu }$$
where *N* is the density of generated free carriers, *I* is the pulse irradiance, hν is the photon energy, *z* is the distance from the focus point along the beam axis (Z axis), *r* is the radial distance from the beam axis, *h* is Planck constant, ν is frequency, and α and $\beta_{2}$ are the linear SPA absorption coefficient and non-linear TPA absorption coefficient, respectively.

The first item on the right hand side of Equation (Equation 1) describes the process of linear absorption (Single Photon Absorption). The irradiance will be attenuated exponentially by the penetration depth, which may give us inconvenience when the beam needs to penetrate deeply into the device [5,21].

In modern IC circuits, there would be many metal layers on top of the transistors; consequently, most of the laser beam would be reflected by these metal layers. A solution for this problem is to apply the laser from back (bottom) side of the chip, so the beam can reach the transistors layer first before reaching the metal layers. The price for this solution is the laser has to travel through the substrate of chips to reach the transistors layer, where the substrate is normally much thicker without any thinning process. Consequently, TPA process is required to accomplish this purpose, which is expressed by the second item on the right hand side of Equation (Equation 1).

In TPA process, it requires ultra-short pulse laser with the pulse duration in femtosecond scale, and the Equation (Equation 1) can be rewritten as:(2)$$\frac{dN(r,z)}{dt}\approx \frac{\beta_{2}I^{2}\left(r,z\right)}{2h\nu }$$

Solve this equation, we can get the following expression.
(3)$$N\left(r,z\right)=T_{pulse\;duration}\times \frac{\beta_{2}}{2h\nu }I^{2}\left(r,z\right)$$

Assuming a Gaussian beam, to get final expression, we can use the following three equations [22].
(4)$$I\left(z,r\right)=I_{0}{\left(\frac{w_{0}}{w}\right)}^{2}exp\left(-\frac{2r^{2}}{w^{2}}\right)$$
(5)$$P_{0}=\frac{1}{2}\left[I_{0}\left(\pi w_{0}^{2}\right)\right]$$
(6)$$w=w_{0}{\left[1+{\left(\frac{z}{z_{0}}\right)}^{2}\right]}^{1/2}$$
where *w* is half the beam diameter, $w_{0}$ is the radius of spot size, *z* is the distance from the focused point, $z_{0}$ is the Rayleigh range, and $I_{0}$ and $P_{0}$ are the maximum beam irradiance and power.

Finally, the expression of N(r,z) is shown like this.
(7)$$N\left(r,z\right)=T_{pulse\;duration}\times \frac{\beta_{2}}{2h\nu }\frac{4P_{0}^{2}}{\pi^{2}w_{0}^{4}{\left[1+{\left(z/z_{0}\right)}^{2}\right]}^{2}}exp\left(-\frac{4r^{2}}{w^{2}}\right)$$

Furthermore, in real testing, we mainly consider the generated free charges around focus point, which means the parameter *z* should be quite close to 0, and $w\approx w_{0}$. Hence, Equation (Equation 7) can be rewritten as Equation (Equation 8).
(8)$$N\left(r,z\right)\approx T_{pulse\;duration}\times \frac{\beta_{2}}{2h\nu }\frac{4P_{0}^{2}}{\pi^{2}w_{0}^{4}}exp\left(-\frac{4r^{2}}{w_{0}^{2}}\right)$$

In the Equation (Equation 8), $\beta_{2}$ is around 1.75 cm/GW with wavelength of 1250 nm [17], hν is very close to 1 eV with wavelength of 1250 nm, $w_{0}$ can be calculated by equation with certain value of Numerical Aperture (NA), *r* can be measured in testing, and $P_{0}$ is a pulse-duration depended measurable power value. By Equation (Equation 8), we can theoretically calculate the generated free charges through TPA process.

To obtain SEEs sensitive mapping by laser system is an interesting researching area. There are some studies related on SEEs mapping for commercial SRAM chips [7,8,9]. However, the dimension of these commercial SRAM cells are mostly several micrometers, and the structures of them are not hardened against radiation [23]. As a result, it is meaningful to do the mapping testing for SRAM cells in a smaller scale and with different structures.

In this study, there are mainly two fields of work. Firstly, a laser system is built up for carrying out the TPA SEES mapping testing. Based on the derived Equation (Equation 8), certain methods and tools are used to measure or monitor key parameters that relate to the generated free charge density *N*. Secondly, after the laser system got prepared, the threshold energy for SRAM cells with different structures will be found; afterwards, the energy distribution profile along Z axis will be achieved, and finally, sensitive mapping will be achieved based on a SEE-hardened SRAM cell. In addition, a comparison between the results from laser mapping and TFIT simulation is also presented by this work.

## 2. Materials and Methods

### 2.1. Ultrafast Laser Testing System Introduction

The laser beam for TPA SEEs testing used in the lab has the wavelength of 1250 nm, and with the pulse duration around 200 femtosecond. Because it’s difficult to get desired pulse laser directly, there are two stages in our laser system that are used to get desired pulse laser. Stage one contains three devices, a seeding laser of Verdi-pumped ultrafast mode-locked Vitesse laser, with Ti:Sapphire as gain medium and fixed 80 MHz repetition rate and 800 nm wavelength; a pump laser of Nd:Vanadate continuous Verdi laser, with 532 nm wavelength and up to 18 W power; an amplifier of RegA 9000 to combine the seeding laser and pump laser together, with a tunable repetition from 10 kHz to 300 kHz and fixed 800 nm wavelength. Eventually, the output pulsed laser from RegA 9000 has the power around 50 mW, repetition of 10 kHz, wavelength of 800 nm, and pulse duration of less than 160 fs. As mentioned earlier, the wavelength for TPA SEEs testing should be larger than 1108 nm, so a second stage is added into our system. There’s only one device in this second stage which is called Optical Parametric Amplifier 9800 (OPA 9800). The OPA 9800 can convert the wavelength in the range from 1200 nm to 1600 nm, with the pulse duration less than 225 fs. All these four devices are manufactured by Coherent, Inc., from Santa Clara, CA 95054 USA. The diagram of laser system is shown in Figure 1.

After pulsed laser came out from OPA9800, a SEEs testing system is built up with such characteristics: (1) ability to monitor the waveform of laser in real time; (2) ability to control the power of laser precisely and repeatably; (3) ability to measure the power of laser in real time; (4) ability to get image of Device Under Testing (DUT); (5) ability to inject the pulse laser into certain region of the DUT; (6) ability to control the number of laser pulses that will be injected into the DUT. Figure 2 illustrates the optical setup of this system.

As shown in the schematic above, a 2:1 beam reducer (could be replaced by a beam-expander) is used to control the beam size in far field. The fast photo-diode (D1) is used to monitor the waveform of pulsed laser. The Attenuator is used to control the power of laser that would pass through it. The Pockels cell is used to temporally control the number of laser pulse that shall be injected into the DUT. The power meter (P1) is used to measure the laser power that will enter the microscope. The Scanner module is used to inject laser beam into certain region of DUT. The Imaging laser is used to obtain the pattern of DUT.

### 2.2. Maintenance and Verification for Testing System

#### 2.2.1. Verification of Pulse Duration

From Equation (Equation 8), the TPA process is highly dependent on the time related parameter of power. Consequently, it is critical to verify the pulse duration. The device for this testing is NT&C Micro$cor autocorrelator from Germany. The measured pulse duration can be seen in Table 1.

In the temporal scale of less than 200 fs, the pulse duration can make the TPA process happen in the focused zone of our laser beam.

#### 2.2.2. Verification of Spot Size

Spot size is a key parameter for laser SEEs testing. Because the profile of laser beam is Gaussian distribution [22], with different spot size, the irradiation would be different correspondingly, and further affect parameters measured in SEEs testing [24,25].

Theoretically, for a Gaussian beam, the minimum waist $w_{0}$ has such relationship with the diverging angle θ, $\theta =2\lambda /(\pi w_{0})$. If *D* is the diameter of the objective lens, *f* is the focal length, then θ≈D/f. Meanwhile, NA=D/2f. So finally, we can get the expression of $w_{0}$.
(9)$$w_{0}=\frac{2\lambda }{\pi NA}$$

In our system, the λ=1.25 μm and *NA* = 0.65, so we can calculate the ideal spot size as 1.225 μm.

If the diffraction limitation is considered with infra-red wavelength, based Airy ring theory, the ideal spot size can be calculated with this equation, $w_{0}=1.22\lambda /NA$, which can get the value of 2.346 μm.

Practically, there are several ways to measure the spot size, either by sharp blade edge or proper scale of grid [26,27]. By blade edge method, the spot size was measured as 2.7 μm, which is closer to the value by Airy ring theory.

Because the profile of laser beam obeys Gaussian distribution, from mathematical calculation [28], by choosing different laser energy, the spatial resolution of the injected laser beam could be changed based on corresponding energy [25].

#### 2.2.3. Verification of Laser Energy

The input energy of pulsed laser, equalling to power/repetition, is very important to SEEs testing. Unfortunately, it’s hard to directly measure the power under the objective lens continuously during our testing. To solve this problem, a power meter (P1, FieldMaxII, Coherent.) behind Pockels Cell is set to continuously record the value of power that is going to enter the Microscope. At the same time, another power meter (P2, PM100USB, ThorLab Inc., from Newton, NJ 07860 USA) is placed under the objective lens to record the value of power that will be injected into the DUT. Apparently, there existing a solid linear relationship between P1 and P2, consequently we can get the value of power under the objective lens (P2) by reading P1 in real time.

To verify the relationship between P1 and P2, we will record a look-up table with corresponding data from P1 and P2. After that, we will keep monitoring the reading from P1 and P2 through whole day. From the data, the output power from laser source would vary in a range, but if we tune the laser source to make the reading from P1 return to a certain value, the corresponding reading from P2 will be back to the same value that is in the look-up table.

To verify the laser energy stability, different amount of energy were injected into the DUT, and recording the corresponding change of generated errors. In following Figure 3, the DUT is a Flip-Flop (FF) chain with BULK technique. Two values of energy (100 nJ and 50 nJ) were injected into a certain FF area in the testing chip, and the related generated number of errors would show the same trend of change: falling into half as the input energy.

The percentage in the brackets represents the waveform noise level of laser pulses. By comparing the error number and noise level, it’s also helpful to find out the trend of error change with different noise level, so as to find a stable working condition for laser system.

In Figure 3, there are five groups of data, with increasing noise level from 12% to 62%. In real testing, if the level of noise could be controlled under 20%, the error number with 50 pJ would be close to half the error number with 100 pJ, which means the laser system works in a stable status; however, as the noise level increased, the total errors generated from both energy declined correspondingly, and the error number with 50 pJ deviates from half the error number with 100 pJ.

#### 2.2.4. Verification of Temporal Control by Pockels Cell

Pockels Cell is a voltage controlled high speed Electro-optical device. Since the repetition of pulsed laser in our testing is 10 kHz, the opening window of Pockels Cell is set to 100 μs. In the real SEEs testing, “Beam scanner module” has the ability to inject a single laser pulse at each step inside a Region Of Interest (ROI) of the DUT. With the setup of Pockels Cell, a certain number of laser pulses could be injected into the ROI area. By this advantage, the injected laser energy can be calculated precisely, and furthermore, this energy can be used to make a comparison with the LET value in heavy ion testing.

#### 2.2.5. Verification of Beam Scanner Module (BSM)

One remarkable characteristic of our testing system is a “Beam Scanner Module”. It is applied to inject the laser pulse into a certain ROI in a DUT, rather than keep the beam still and move the stage to scan the ROI. BSM system consists of a pair of high reflective mirrors, each of which is controlled by voltage signal. The BSM system has the following advantages compared with the “stage moving system”:it moves laser beam with high velocity while scanning the ROI;it moves the laser beam precisely inside the ROI;Since the stage is still, there is no physical vibration applied to the DUT.

A SRAM chip can be used to verify the performance of BSM. As shown in Figure 4, there is a SRAM block named “Regular_11T”. It has 256 SRAM words (each unit has 8 SRAM cells inside) in this block with the allocation of 64 (row) by 4 (column). Furthermore, each SRAM word has a unique physical address, and if there is an error generated inside a word, the address of that word will be recorded automatically. In the verification, laser pulse was shot into the words of (1,0), (1,1), (2,0), and (2,1). The results of corresponding address and generated errors is shown in Table 2. From the results, we can see the address difference between words (1,0) and (2,0) is 1, whereas the address difference between words (1,0) and (1,1) is 64. It can verify our BSM can inject the laser beam precisely into the DUT.

### 2.3. SRAM Chip in the Testing

In the SRAM testing chip, there are five SRAM blocks with different structures, which are traditional_6T, Layout Design Through Error Aware Transistor Positioning_11T (LEAP_11T), Quatro_10T, Regular_11T, and Proposed_6T. The block diagram and die image is shown in Figure 5.

## 3. Results and Discussion

### 3.1. Threshold Energy for Generating Errors in Different Structure of SRAM

Threshold energy has been measured for different SRAM structures, which are traditional 6T, LEAP 11T, Quatro 10T, and Regular 11T, respectively. From the view of circuit structure, traditional 6T is the most vulnerable structure to radiation for it just uses two cross-coupled inverters to store bits information; the Regular 11T uses Cascode Voltage Switching Logic (CVSL) structure, and uses four complementary nodes to stand for two logic states, which gives this structure an ability of error correction; LEAP 11T is a modified design for regular 11T. It didn’t change the structure of original design, but re-located the position of PMOS and NMOS so as to make the collected charge eliminated by themselves; finally, the Quatro 10T has a symmetrical structure that has the best ability of anti-radiation by turning off certain transistor to maintain original status [29].

In threshold energy testing with data pattern of all 0 in SRAM, laser pulse with the energy from 10 pJ to 100 pJ was injected into a single SRAM cell in each SRAM structure, by a step of 10 pJ. Once the threshold energy to generate error was found, repeated verification testing would be fulfilled for three times. Finally, the result is shown in Table 3, where the Quatro structure is the most non-sensitive design, whereas the Traditional 6T structure is the most sensitive design.

For comparison, a result from former Alpha radiation testing to the same SRAM chip is also shown in Table 3 [30]. From the result, we can clearly see the sensitivity for each structure. With the same applied voltage (chip driven voltage), traditional 6T is the most sensitive structure than the other three, whereas the quatro 10T is always the most non-sensitive structure; meanwhile, the sensitivity of regular 11T and LEAP 11T locates between those two structure, and the regular 11T is the same sensitive with LEAP 11T.

Finally, the Equation (Equation 8) can be used to calculate a theoretical number of free charge generated by certain threshold energy for each single laser pulse. For calculation, the $T_{pulse\;duration}\approx 200\times 10^{-15}$ s, while r=2.7 μm and $w_{0}=1.225$ μm. The value of $P_{0}$ can be achieved by each threshold energy. Table 3 shows the theoretical number of free charge in column 4.

The results from Alpha testing are coherent with laser testing results. The most sensitive design of Traditional 6T has the lowest threshold energy of 30 pJ, whereas the most non-sensitive design of Quatro 10T has the highest threshold energy of 70 pJ. The threshold energy of both Regular 11T and LEAP 11T has the same value of 50 pJ. This phenomenon is reasonable because they have the same structure, and from the result of Alpha testing, the sensitivity for Regular 11T and LEAP 11T is also the same.

### 3.2. Energy Distribution along Z Axis

In the analysis in Part 1, from Equation (Equation 8), the generation of charges is highly dependent on injected laser power. Consequently, it’s important to research the distribution profile of laser energy along Z axis, so that an active zone of the focused laser beam can be achieved.

From Equation (Equation 4), if the focus length **r** is within the Rayleigh range $z_{0}$, we can assume the beam radius *w* is equal to the focused beam radius $w_{0}$, so Equation (Equation 4) can be rewritten as:(10)$$I\left(z,r\right)=I_{0}\;exp\left(-\frac{2r^{2}}{w_{0}^{2}}\right)$$
where *z* means a point on beam axis, $I_{0}$ is focused beam irradiance, $w_{0}$ is the focused beam radius.

Here, we let $I=\frac{1}{e}I_{0}$, and $w_{0}$ is measured as 2.7 μm. Hence, we can solve Equation (Equation 10), and get the value of *r* as 1.91 μm. It means the focused zone along z axis will be ±1.91 μm.

In testing, at the beginning, the laser beam is focused at the best position with the threshold energy. After that, we moved the beam upwards with the step size of 1 μm. At each position, the threshold energy would be recorded to monitor the change of energy. When this work was finished, we moved the beam downwards from the best focused position and repeated the same procedure. The data is shown in Table 4.

From the results, the active zone along z axis is from −3 μm to +2 μm, which is quite close to the calculated value of ±1.91 μm.

From the data, if we convert the relationship between energy and generated errors, we can get a laser beam profile along z axis, which is shown in Figure 6.

### 3.3. Sensitive Map and Simulation for Quatro Structure of SRAM

A specific system is developed to do the mapping testing. Three hardware is involved: a desktop is used to control the laser and generate sensitive map; a raspberry pi is used to control the FPGA and generate SEEs error information; a FPGA board and SRAM daughter card are used to generate output data. The total data flow is shown in Figure 7.

#### 3.3.1. Sensitive Map of Quatro Structure

The Quatro structure of SRAM was firstly proposed in 2009, which contains 10 transistors (6 NMOS and 4 PMOS) and has an ability of self soft error correction [29]. The diagram and layout of this structure are shown in Figure 8.

Firstly, the stored data pattern was set to all ‘0’. From Figure 8a, the nodes of ‘A’, ‘B’, ‘C’ and ‘D’ should be ‘0’, ‘1’, ‘1’, and ‘0’.

If the transistor N3 is struck by a laser pulse, the voltage of its drain (node ‘B’) would drop, and as a result, the transistors N1 and N4 may be turned off. This may lead to a cascade of pulling up and flipping the voltage of node ‘D’, furthermore pulling down and flipping the voltage of node ‘C’, which would eventually change the voltage of node ‘A’ from ‘0’ to ‘1’, and generate an error.

The related sensitive map for this case is shown in Figure 9. The dimension of whole SRAM Quatro cell is 1.10 μm by 1.80 μm, and the step size of scanning is 0.18 μm.

From the circuitry analysis, in this sensitive map, we can observe the area of node ‘B’ turns red, which means it’s the sensitive area with this setup, which is consistent with theory.

Secondly, the stored data pattern was reset to all ‘1’. Consequently, the nodes of ‘A’, ‘B’, ‘C’ and ‘D’ should be ‘1’, ‘0’, ‘0’, and ‘1’.

By this setup, if the transistor N1 is struck by a laser pulse, the voltage of its drain (node ‘A’) would drop, and as a result, the transistors N2 and N3 may be turned off. This may lead to a cascade of pulling down and flipping the voltage of node ‘D’, and correspondingly pulling up and flipping the voltage of node ‘C’. Eventually an error will be generated in the circuit.

The related sensitive map for this case is shown in Figure 10, with the same dimension as before.

Because the stored data is changed to all ‘1’, the sensitive area is switched to the right part and covered the node ‘A’ area, which is also consistent with our analysis.

#### 3.3.2. TFIT Simulation of Quatro Structure

TFIT is a common used commercial simulation tool based on SPICE netlist and chip layout. It can generate a simulation for sensitive area in a certain device according to different energy level (LET value).

In order to examine the validity of the laser SEU mapping results, 10T Quatro cell simulation has been conducted on the 65 nm CMOS process model by using IROC TFIT tool.The similar device model is created and calibrated in previous [31,32,33]. As shown in Figure 11, TFIT simulation results are obtained with the LET from 0.01 to 0.50 pC/μm for “00H” and “FFH” data pattern respectively.

The background of the SEEs map is the active region of the memory cell and the locations of upsets induced by ion strike are identified by colored squares. The color corresponds to the LET value. The TFIT simulation results show that the drain areas of OFF driver NMOS (node A/node B) and OFF pull-up PMOS (node C/node D) are sensitive to SEU for “FFH” and “00H” data pattern respectively. The SEU sensitive regions are well consistent with the laser SEU sensitivity mapping results and previous work. Meanwhile, it notes that the sensitive area obtained in mapping results with threshold laser energy is much larger than that obtained in simulation results due to the laser spot size and charge carrier diffusions [24]. It is shown in Figure 9 and Figure 10 that the resolution of laser SEU sensitivity mapping technique is about 0.5 μm, indicating that laser mapping technology is reliable for this deep submicron technology device.

From the simulation, we can see the most sensitive area (red area) is located on the “Node B” with all ‘0’ data pattern, and “Node A” with all ‘1’ data pattern, respectively. The simulation results are consistent with our testing results.

## 4. Conclusions

TPA laser facility has been widely used for SEEs testing in recent years; however, there are still several uncertainties in the practical testing. This paper derives an equation that can predict the generated number of charges by TPA procedure. After that, several verifications have been done to give us clear practical methods to control the system in an optimal working status.

Three meaningful testing was carried out sooner. Firstly, the threshold laser energy for different SRAM structures has been confirmed by the data from former Alpha testing, which proves the energy resolution of the laser system. After this, the energy distribution along z axis was achieved, and the data is compared with theoretical data, which is quite close. This result will give us much inspiration in practical TPA laser testing. Finally, a sensitive map based on SRAM Quatro structure was achieved by a real-time mapping system. The generated sensitive area is consistent with the theoretical analysis, and is further verified by TFIT simulation results. Besides, the resolution of mapping can be achieved around 500 nm, which gives us a useful way to research the sensitivity inside a device with micron-metre scale.

In the future, more work will be carried out with the purpose to decrease the spot size, and modify the spatial resolution of BSM in our laser system. By decreasing the spot size, a smaller resolution can be gotten and it will help us to reach present 28 nm devices or with even smaller size. By modifying BSM, we can control the movement of our laser beam in a more precise way.

## Figures and Tables

**Figure 1 materials-12-03411-f001:**
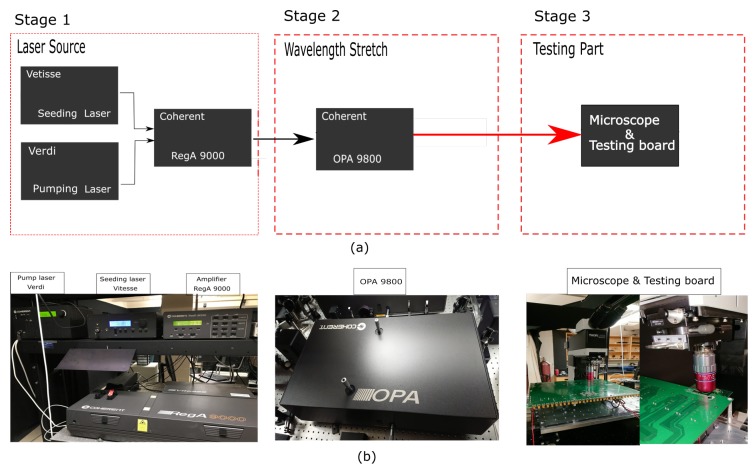
(**a**) Schematic of SEEs testing laser system. (**b**) Real profile of laser system.

**Figure 2 materials-12-03411-f002:**
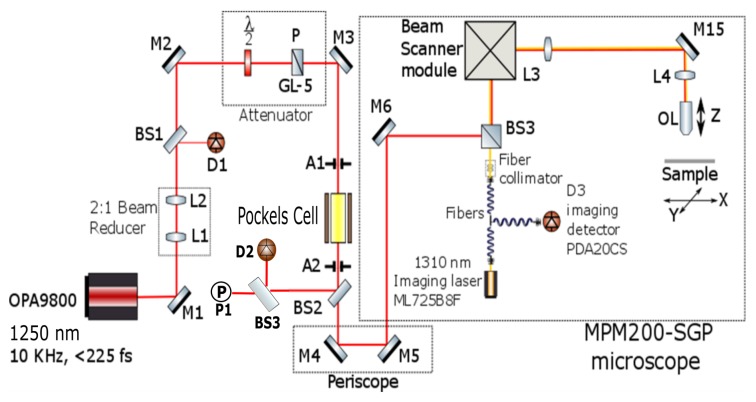
Schematic of setup for SEEs mapping testing after OPA9800.

**Figure 3 materials-12-03411-f003:**
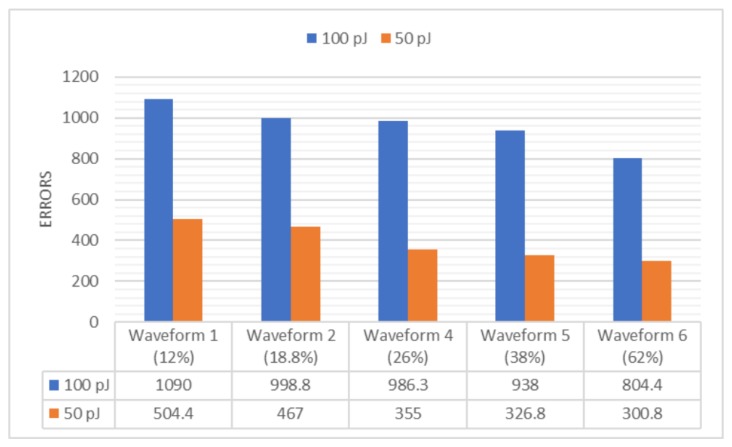
Laser energy stability: the change of error numbers (average value from 10 rounds) with the change of laser energy and noise level.

**Figure 4 materials-12-03411-f004:**
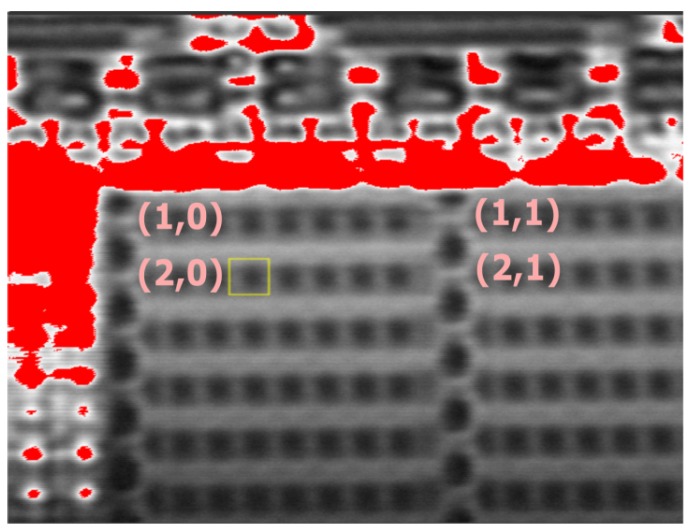
Schematic of SRAM words in a SRAM block.

**Figure 5 materials-12-03411-f005:**
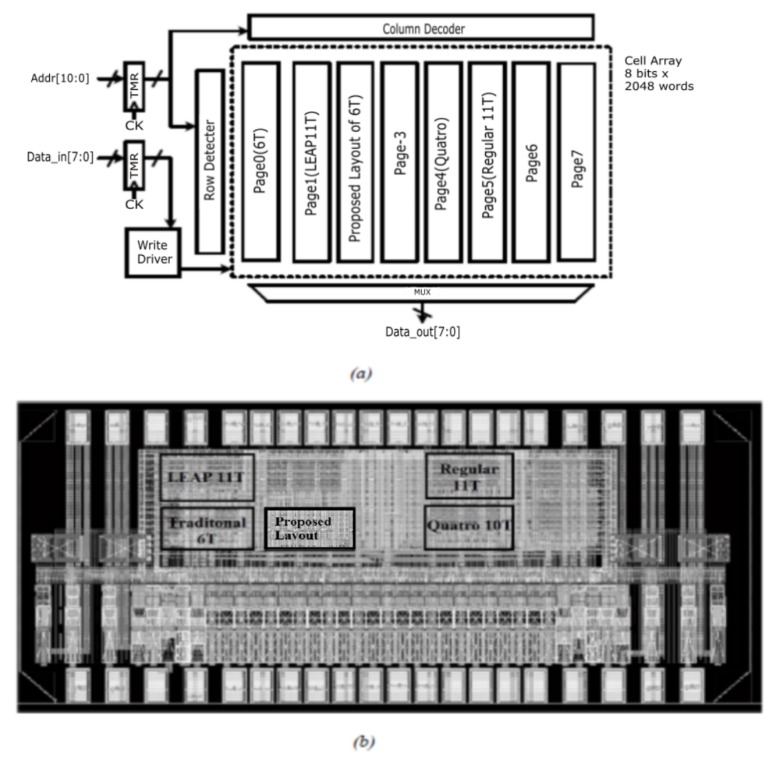
(**a**) Block diagram of SRAM chip; (**b**) die image of SRAM chip.

**Figure 6 materials-12-03411-f006:**
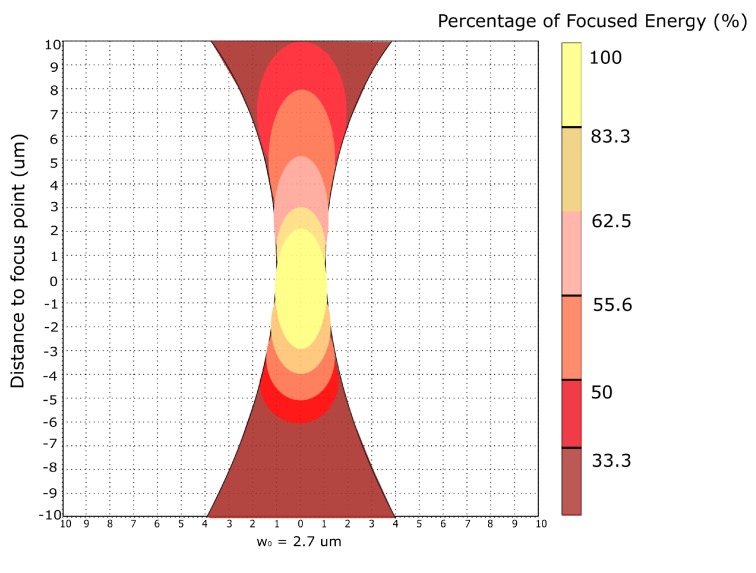
Energy distribution profile of focused laser beam.

**Figure 7 materials-12-03411-f007:**
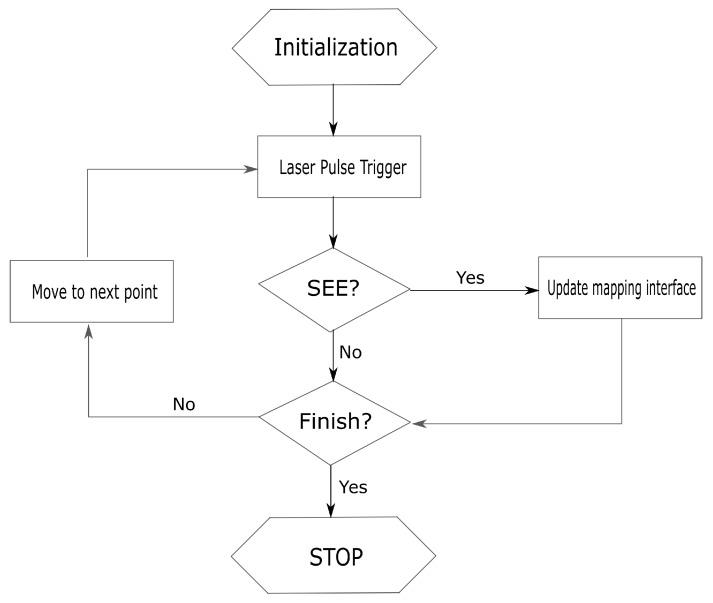
Data flow of whole mapping system.

**Figure 8 materials-12-03411-f008:**
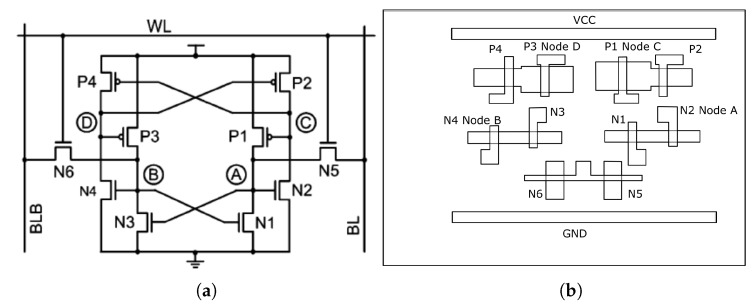
(**a**) Diagram of Quatro design [29]; (**b**) Layout of Quatro design.

**Figure 9 materials-12-03411-f009:**
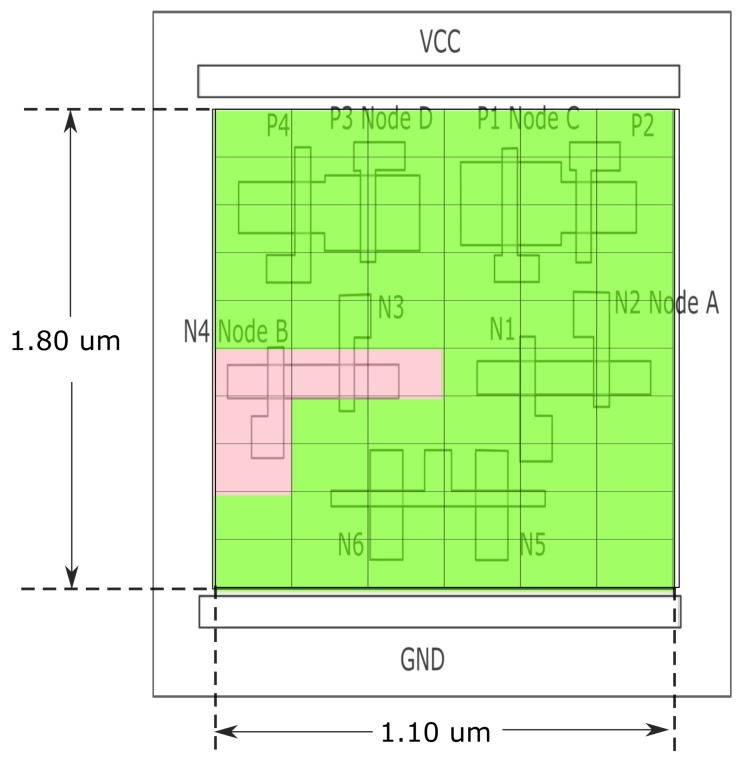
Sensitive map for Quatro design of SRAM with all ‘0’ stored data pattern and 90 pJ laser energy.

**Figure 10 materials-12-03411-f010:**
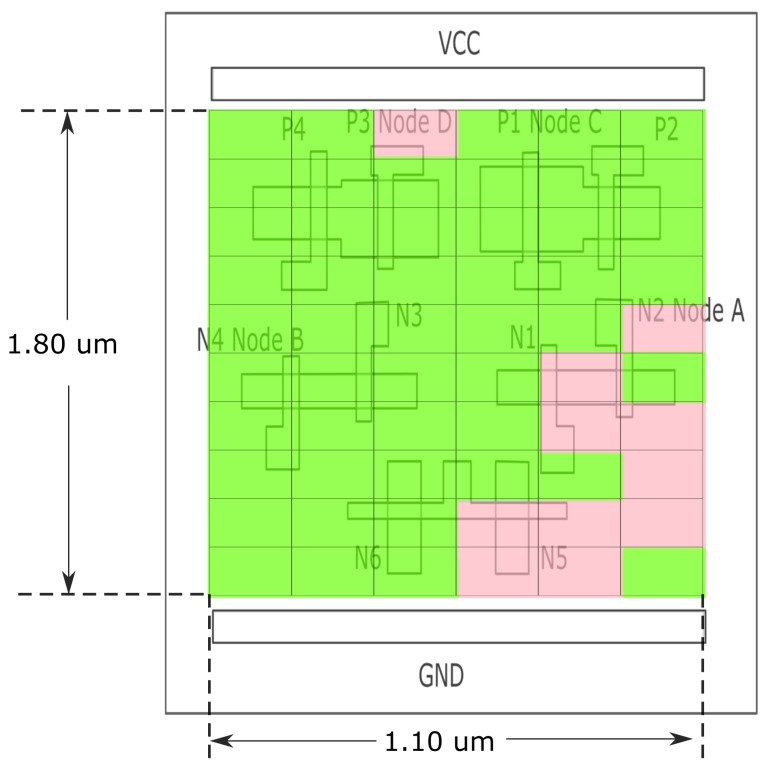
Sensitive map for Quatro design of SRAM with all ‘1’ stored data pattern and 90 pJ laser energy.

**Figure 11 materials-12-03411-f011:**
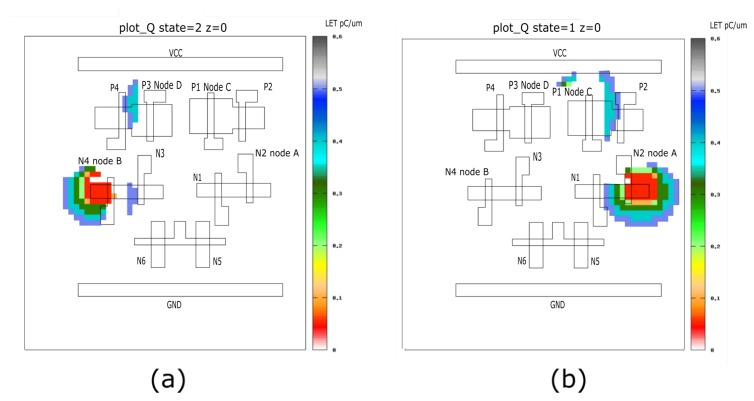
(**a**) TFIT simulation sensitive area with all ‘0’ stored data pattern (**b**) TFIT simulation sensitive area with all ‘1’ stored data pattern.

**Table 1 materials-12-03411-t001:** Measured value of pulse duration for each laser device.

	Vitesse	RegA 9000	OPA 9800
Pulse Duration (fs)	∼95	∼140	∼200

**Table 2 materials-12-03411-t002:** Generated errors and corresponding address from a SRAM block.

Location	Energy (pJ)	Address	Errors
(1,0)	80	1038	1
(1,1)	80	1102	1
(2,0)	80	1039	1
(2,1)	80	1104	1

**Table 3 materials-12-03411-t003:** Comparison between Laser Threshold Energy and Alpha Radiation Error rate for each SRAM structure.

SRAM Structure	Threshold Energy (pJ)	Error Rate of Pattern All 0 (%)	Number of Free Charge for Single Pulse (/cm^3^)
Traditional 6T	30	0.012	$2.0\times 10^{14}$
Regular 11T	50	0.001	$3.4\times 10^{14}$
LEAP 11T	50	0.001	$3.4\times 10^{14}$
Quatro 10T	70	0.0005	$4.76\times 10^{14}$

**Table 4 materials-12-03411-t004:** Laser threshold energy distribution along Z axis.

Position Upwards (μm)	Threshold Energy (pJ)	Position Downwards (μm)	Threshold Energy (pJ)
0	50	0	50
+1	50	−1	50
+2	50	−2	50
+3	60	−3	50
+4	80	−4	70
+5	80	−5	100
+6	90	−6	110
+7	100		
+8	100

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
