# Peer review of "Application of Two-Photon-Absorption Pulsed Laser for Single-Event-Effects Sensitivity Mapping Technology"

_materials, 2019, doi:10.3390/ma12203411_

Round 1

Reviewer 1 Report

The manuscript (#materials-604178) describes a laser system on the SEE sensitivity mapping of SRAMs. After carefully reading the manuscript, however, I cannot recommend the manuscript for publication.

The authors provided the details of the laser system in the section of Materials and Methods and its capability on SEE sensitivity mapping. However, it has been known in a bunch of literature of this field. In the section of Results and Discussions, the authors showed a data flow of the mapping system, but no test result of sensitivity mapping on SRAMs was shown in the section. Only simulation results were shown.

The manuscript did not meet the requirement of a technical paper. Therefore, I cannot recommend the publication of the manuscript.

Reviewer 2 Report

The subject is interesting, and the manuscript is well organized. However, for better clarity, I suggest a change in Figure 7. There is no value in the horizontal axis. The distance is proportional to the values on the vertical axis, however it worth to add the values for the reader's convenience. The “w” (beam waist) should be also defined in the caption of the horizontal axis.

Reviewer 3 Report

The Application of Two-Photon-Absorption Pulsed Laser for Single-Event-Effects Sensitivity Mapping

MDPI Materials Sep. 2019

In this paper Single-Event-Effects Sensitivity Mapping has been carried out on SRAM using the Two-Photon-Absorption process in the silicon semiconductor active region. The proposed test-setup works nice, and experimental results show the applicability of this test method.

Title: remove The…..

Abstract: Z axis?? Define that… instead e.g. write in the axial direction…

Page 2 line 35: 1.108 um, remove one digit: 1,11 um

Page 2 line 35: ….the injected laser light……

Page 2 line 44-45: ….in a semiconductor material….

Page 2 line 47: ….the linear and non linear TPA…coefficients….

Equation 2: write an approximation when the TP process dominates..

Equation 3: how has is the time dependence been removed?

Page 2 line 62: Assuming a Gaussian beam…

Equation 6: w, w0 and omega looks quite similar!!

Equation 8: write approx. and for r->0

Page 3 line 83: Z axis ??

Page 3 line 97: Why exactly 1250 nm wavelength and not eg. 1300 nm?

Figure 1: add more figure test description (the is the same for almost all figures!)

Should stage 1 not contain a pump?

What is the SPA output used for??

Figure 2: please show the correct direction of z with only one arrow

Page 5 line 118: Remove “once a week” amd in line 144 remove “in the morning”

Page 5 line 126: This equation is NOT valid for e Gaussian beam!!!! This is the equation valid for diffraction through a circular aperture!! What is the beam diameter before entering the microscope objective compared to the entrance diameter??

Figure 3: the interpretation of this figure is not clear, please update text accordingly.

Page 6 line 158: kHz not KHz (also a mistake other places in the manus)

Figure 4 is not giving any information, can be written using only text. And, if the figure should be used, convert data to a nice graph in a suitable graph program and not a screen-shot from an oscilloscope.

Figure 5; what does the red color indicate??

Figure 6: change to “proposed”

Page 9 line 203: Is alpha radiation the same as Heavy Ion (HI) testing?. If yes, use the same nomenclature.

Merge table 3 and 4 for direct comparison.

Figure 7: is this a calculated plot??

What info is gained from this simple plot – it is not clear for what purpose it has been used in the obtained measurement results. I think this figure can be removed.

Figure 10: explain color scale. One could also using a fine mesh indicate the step size of 0.18 um , in total 60 scan-points.

What is the speed of this scanning TPA method?

Figure 11: explain color scale, grid,….

Figure 12: text is to small. Indicate node A and node B. Explain color scale.

Page 14 line 285: eq. 8 has not been used to correlate N with puls energy. This is an important information!!

Reviewer 4 Report

This manuscript reported a customized laser system for carrying out 80 the TPA SEES mapping testing. In addition, a comparison between the results from laser mapping and TFIT simulation is also presented by this work. The manuscript is clearly structured in Introduction, Analysis, Conclusion and Bibliography. The text is comprehensible and concentrates on the essential results. The abstract presents the essential statements of the article in a logical context.

This paper is worth of publication in the Journal of Materials, if the following changes/revisions can be applied/addressed (major revision):

The results have been clearly shown; however, the manuscript suffers from lack of sufficient discussion for the presented results. The novelty of the manuscript is not clear; more info about the similar research projects in the filed can be added. Authors should explain about the stats methods used in this research; Also, number of samples/tests and errors are missed. Font size in some images should be increased (e.g. Figure 6)
